# Mechanisms Involved in the Persistence of *Babesia canis* Infection in Dogs

**DOI:** 10.3390/pathogens8030094

**Published:** 2019-06-29

**Authors:** Theo Schetters

**Affiliations:** 1ProtActivity R&D, 5431 Cuijk, The Netherlands; th.schetters@protactivity.com or; 2Department of Veterinary Tropical Diseases, University of Pretoria, Onderstepoort 0110, South Africa

**Keywords:** Babesia canis, sequestration, blocked capillaries, shock, coagulation, inflammation

## Abstract

Dogs that are infected with *Babesia canis* parasites usually show severe clinical signs, yet often very few parasites are detectable in the blood circulation. The results showed that large numbers of *B. canis*-infected red blood cells accumulate in the microvasculature of infected subjects. The initial process leading to the attachment of infected erythrocytes to the endothelial cells of small capillaries (sequestration) appears to involve the interaction of parasite molecules at the erythrocyte surface with ligands on the endothelial cells. Since parasites continue to develop in the sequestered erythrocyte, it would be expected that the infected erythrocyte is destroyed when the mature parasites escape the host cell, which would make it hard to explain accumulation of infected erythrocytes at the initial site of attachment. Apparently, additional processes are triggered that lead to consolidation of parasite sequestration. One possible explanation is that after initial attachment of an infected erythrocyte to the wall of a blood capillary, the coagulation system is involved in the trapping of infected and uninfected erythrocytes. The data further suggest that newly formed parasites subsequently infect normal red blood cells that are also trapped in the capillary, which finally leads to capillaries that appear to be loaded with infected erythrocytes.

## 1. Introduction

Dogs that are infected with *Babesia* parasites present a wide range of clinical manifestations, varying from circulatory shock to multi-organ dysfunction and additional complications [1,2,3]. Little is known why the disease manifests itself differently in individual dogs. There are some indications that *Babesia* parasites vary with regard to virulence, both at the species and isolate level [4,5,6]. In addition, the outcome of infection appears to be also dependent on the number of parasites that infects the animal [7,8]. Many of these parameters are unknown when dogs are presented at the small animal practice. Even the time of infection can often only be guessed from circumstantial evidence, such as the day animals were walked in tick-infested areas or when a tick was recovered from the dog. Unravelling the sequence of events that leads to clinical disease is almost impossible from field cases, and results from the clinical practice usually pertain more to the agonal stage of disease. Controlled experimental infection with defined parasite isolates allows the study of the early events of *Babesia* infection and subsequent development of clinical disease. 

## 2. Experimental Infection Models

The natural route of infection with *Babesia* parasites is by tick bite, although dogs can also become infected during fighting, and it has even been suggested that dogs may become infected by ingestion of an infected tick [7]. When dogs are experimentally infected by a tick bite, relatively few parasites are transmitted as deduced from the time period before parasites become detectable in the peripheral blood, which varies from 8–21 days (average 14 days; [7,9]). Alternatively, dogs can be infected by injection of infected erythrocytes either from blood of *B. canis*-infected dogs or from *in vitro* cultures of the parasite. Several research groups that have studied canine babesiosis report that the disease is essentially similar irrespective of the method of infection [7,10]. During the development of a commercial *B. canis* vaccine over the last three decades [11], the author used a standardized infection model in which dogs of approximately 6 months of age were infected with heparinised blood of a dog that had been infected with 1 ml of *B. canis* stabilate that was stored in liquid nitrogen [4,12] This procedure allowed the establishment of *Babesia* infection reproducibly. The data reviewed here are obtained from this model, unless otherwise indicated.

## 3. Blood Circulation Problems

Dogs that were infected with blood from a *B. canis*-infected dog typically showed lethargy, anorexia and pale to white mucosa (Figure 1; [13,14]). In addition, the capillary refill time was increased and the pulse was sharp. These signs indicated that there was impairment of the blood circulation and reduced tissue perfusion. Indeed, histological examination of the mucosae of a dog with pale-white mucosae showed the absence of erythrocytes in the capillaries, whereas capillaries of a non-infected control dog were filled with blood (Figure 1). In contrast to the peripheral tissues, the internal organs of *Babesia*-infected dogs were heavily loaded with blood. It appeared that the red blood cell concentration at the venous side of the spleen was significantly increased as compared to the concentration at the arterial side, which is indicative of congestion and results in splenomegaly [15]. Further histological examination of dogs that developed severe clinical symptoms showed that some capillaries of internal organs appeared to be blocked with cellular material entrapped in proteinaceous material. The parasitaemia in those capillaries reached 100% (Figure 2). These findings are similar to those reported by Graham-Smith [16] who evaluated the parasitaemia in blood and different organs from eleven dogs that died from babesiosis. Although parasitaemia in the blood circulation was on average 1.5% (range 0.1–6.0%), the parasitaemia in the capillaries of internal organs was much higher and varied from 12–95%. It was further noted that in most organs, the capillaries were dilated, which confirmed observations of Nocard and Motas [13] (Table 1).

Blocking of the capillary bed could lead to poor tissue perfusion. However, in the same section of different organs, it appears that some capillaries are affected whilst others are apparently normal (Figure 3). This indicates that blocking of the capillaries is not a generalized phenomenon, but the result of a process that is localized to the affected capillary, instead. The data suggest that there is some stochastic event involved. The question is, what is the initial event that leads to accumulation of (infected) erythrocytes in the capillaries? 

## 4. Mechanisms of Obstruction of Capillaries

In blood smears of *B. canis*-infected dogs, the infected erythrocytes are often found in small agglutinates. These agglutinates of infected erythrocytes have also been observed in capillaries on histological sections of *B. canis*-infected dogs (Figure 4). Apparently, the infected erythrocyte surface is altered such that there is an affinity between the cells, strong enough to survive hydrodynamic forces in the circulating blood. The blocking of small capillaries may result from entrapment of such agglutinates of infected red blood cells. It is intriguing from where these agglutinates arise. Given the fact that the percentage of infected red blood cells is usually much less than 1% [8], the odds that agglutinates form because of accidental encounter of two or more *Babesia*-infected erythrocytes in the circulating blood seem too low to explain this. It is more likely that these agglutinates were formed locally before entering the blood circulation and subsequently disseminated to other capillaries.

## 5. Babesia Infection Induces Hypotension

The pathophysiology of *B. bovis* has been extensively studied by the group of Ian Wright who suggested that the initial event was hypotension, a phenomenon that was also found in dogs infected with *B. rossi* [17,18]. In order to study this, the blood pressure was determined in groups of dogs that were infected with graded doses of *B. canis*-infected erythrocytes. The results showed that the mean arterial blood pressure (MAP) started decreasing already two days after infection with 1 × 10^6^
*B. canis*-infected erythrocytes. In dogs that had been infected with a 100-fold lower dose, the onset of hypotension was delayed by two days (Figure 5; [8]). The blood pressure of the dogs that were infected with the highest dose was restored the day after treatment. Importantly, the blood pressure of the dogs that had been infected with the 100-fold lower dose, restored within two days spontaneously, without chemotherapeutic treatment. 

The results from continuous *in vitro* cultures of *B. canis* showed that parasites divide by binary fission and that the proliferation factor of *B. canis* in 24 h was 11.0 (±3.1; Figure 6). Hence, the fact that the onset of hypotension in experimentally infected dogs was delayed by two days when 100-fold fewer parasites were used for infection could be explained, assuming that a certain threshold number of parasites were required to trigger this phenomenon. Within two days, the number of infected erythrocytes would theoretically be increased 100-fold.

## 6. Compensated Hypotension

In malaria, hypotension leads to increase of blood volume to restore blood pressure, which results in haemodilution [19,20,21]. To further compensate for reduced oxygen transport due to hypotension, a number of additional mechanisms are triggered. The heart rate is increased through the sympathetic nervous system (to increase cardiac output), and the respiration rate is increased [21]. Similar symptoms have been reported as early as 1904 in dogs suffering from babesiosis by Nuttall who stated: “The breathing and pulse are accelerated, the former becomes laboured and finally shallow” [10]. Chemotherapeutic treatment of infected subjects leads to restoration of normal blood volume (increased urination) and capillary resistance [19,22]. This haemodynamic state is called compensated hypotension [21]. 

The experimental *B. canis*-dog model studied here allowed to estimate some parameters associated with compensated hypotension. Dogs that that were infected with 1 x 10^6^
*B. canis*-infected erythrocytes became thirsty and produced less urine from day 2 after infection onwards, hence they retained water. This was reflected in the body weight, which increased with 10% in three days’ time (Figure 7; [23]).

In order to determine the effect on the blood volume, a parameter that is naturally present at a relatively stable concentration in the blood was selected. If the blood volume was increased, the concentration of this compound would be decreased. The total protein concentration in the plasma initially decreased (data not shown), but because of acute phase responses and the induction of immunoglobulins, which affect the protein concentration, this parameter was not used. Instead, creatinin, which is a metabolite from muscle that is produced at a constant rate and excreted by the kidney, was used. The results showed that in early stage *B. canis* infections, when there is no evidence of kidney damage, the concentration of creatinin in the plasma decreased with time at the same rate as the red blood cell concentration (Figure 8; [8,15,23]). Similar reduction of the creatinin concentration in another model of early experimental *Babesia* infection in dogs has been reported more recently [24]. Interestingly, the dynamics of PCV decreases after experimental *B. canis* infection is remarkably similar in different publications, whether these come from very old reports or more recent. This has been interpreted that this is a physiological response that is switched on or off (reviewed in [11]). The time point that this switch is operated depends on the moment the parasite load has reached a certain threshold in the infected dog, which is in part dependent on the parasite isolate used and the number of parasites in the infectious inoculum. It should be realised that this parameter cannot be used in more advanced *Babesia* infections in dogs when there is clear evidence of kidney damage, in which case the creatinin concentration has often been found to be increased in the plasma [1,2].

In conclusion, at this phase of infection, the blood pressure appears normal but is actually a compensated state with decreased capillary resistance. Clinically, in the *B. canis*-infected dog, this is evidenced by prolonged capillary refill time and a sharp pulse. In malaria, this is known as orthostatic hypotension, which is characterised by the fact that patients become dizzy and may faint when standing up because of venous pooling of the blood in the lower limbs due to decreased vascular resistance [20]. In such a case, also the velocity of the blood in the capillary bed is decreased, which may lead to stasis [25].

## 7. Acute Phase Response-Coagulation

Infection of dogs with *B. canis*-infected erythrocytes triggers the acute phase response as seen in other systemic infections [8,21]. At the second day after infection of the dogs with 1–2 × 10^6^
*B. canis*-infected erythrocytes, the acute phase protein C-reactive protein (CRP) was markedly increased, and further, increased fibrinogen levels were found one day later. The moment that parasitaemia decreases (after reaching a peak), it coincides with a marked decrease of plasma fibrinogen levels and a simultaneous increase of APTT times, which is indicative of consumptive coagulopathy [4]. This suggests that coagulation plays a role in the decrease of parasitaemia in the circulating blood (see Figure 9).

In order to find evidence for this, the authors treated a *B. canis*-infected dog that had become parasite-negative in blood smears with heparin. Two days after the heparin injection, parasites were detected in blood smears, and when parasitaemia decreased again, subsequent injections with heparin were again followed by a transient rise in parasitaemia (Figure 10; [23]). Interestingly, these transient increases were associated with transient increases of PCV (number of erythrocytes in the circulating blood). The fact that the percentage of infected erythrocytes in the circulation increased after heparin treatment suggest that sequestered *B. canis*-infected erythrocytes are released into the circulating blood. 

The results from the histological sections from a dog that was infected with *B. canis* showed that infected erythrocytes were located in fibrin deposits at the margin of the capillary. In the centre of the capillary, normal erythrocytes were found that were not associated with fibrin (Figure 11). This histological section shows that a section with a capillary that is packed with infected erythrocytes can be found, which could lead to the conclusion that the entire capillary is blocked (as in Figure 2). However, here it is shown that might not be correct, and that there could still be blood flow in the central part of the capillary (see also Figure 9). Margination of infected erythrocytes in the capillaries of infected dogs has been observed before by Graham-Smith in 1905 who stated that: ‘’From sections it was seen that in all cases the smaller capillaries contained more numerous infected corpuscles than the larger vessels. In the latter, the infected corpuscles were generally found near the wall of the vessel” [16]. More recently, it was found that in natural *B. rossi* infection in dogs mortality was associated with consumptive coagulopathy. This suggests that this phenomenon could be more wide spread [26]. 

The fact that relatively large numbers of infected erythrocytes are found in fibrin deposits could explain why the injection of heparin increased the parasitaemia in the circulating blood. By inhibiting the formation of fibrin, the balance between clot formation (due to fibrin formation from fibrinogen) and clot resolution (due to fibrin degradation by plasmin) is shifted towards clot resolution. This would result in releasing infected erythrocytes in the blood circulation. Alternatively, the injection of heparin could have interfered with the process of sequestration e.g., by inhibition of receptor-ligand interaction between infected erythrocytes and endothelial cells. More research is required to elucidate the exact mechanisms.

## 8. Parasite Sequestration

The fact that in the circulating blood the proportion of erythrocytes that harbours a single *Babesia* parasite outnumbers the proportion of erythrocytes with two parasites (that is at least 8 h infected because one round of multiplication has been completed) suggests that older infected erythrocytes are removed from the circulation (Table 1). This could be due to (innate) immune mechanisms that actually destroy the infected erythrocytes, but it could also be due to sequestration in the internal organs, or a combination of both mechanisms. 

It has been suggested that in babesiosis, as in *Plasmodium falciparum* malaria, protrusions at the infected erythrocyte surface are associated with adherence of infected erythrocytes to the endothelium of blood vessels, a process called sequestration [27]. In *B. bovis*-infected erythrocytes such structures are evident, but absent on the erythrocytes that are infected with the *B. canis* isolate used here (Figure 12). 

In *B. bovis,* these structures are sites where specific parasite molecules are found that bind to ligands on endothelial cells. The most important appear to be the variant erythrocyte surface antigens (VESA; [28]). The fact that there are no knob-like protrusions on *B. canis*-infected erythrocytes does not exclude the possibility that molecules at the erythrocyte surface are involved in the binding to endothelial cells. Indeed, VESA-like genes have been described in the *B. canis* genome [29], and live immunofluorescence using serum from dogs that were immune to re-infection showed the presence of species-specific parasite antigens on the surface of *B. canis*-infected erythrocytes (Figure 13; [30].

In conclusion, *B. canis*-infected erythrocytes do not exhibit knob-like structures, but they do express parasite antigens at the red blood cell surface. These antigens could play a role in the initial attachment of infected erythrocytes at the endothelial cells of blood vessels (margination). More research is required to elucidate which parasite molecules are expressed.

## 9. Local Proliferation

The fact that most, if not all, erythrocytes that are trapped in the fibrin mass are infected is intriguing. It is difficult to understand that infected erythrocytes would accumulate in specific capillaries, especially since the majority of parasites escape the erythrocytes when they proliferate, and invade new host cells. This led us to hypothesize that infected erythrocytes that adhere to the endothelial cells are trapped together with uninfected erythrocytes in fibrin deposits where they continue to proliferate [31]. This leads to plaques with almost all erythrocytes being infected. The presence of small agglutinates on *B. canis*-infected erythrocytes that are observed in blood smears and in the circulation of *B. canis*-infected dogs (Figure 4) could result from such areas where parasitaemia reaches high values. These agglutinates could subsequently block additional capillaries more easily and trigger further local inflammatory reactions.

In his detailed analysis, Graham-Smith also counted the number of free *Babesia* parasites per infected erythrocyte (Table 1). The data showed that the number of free parasites was much higher in the capillaries than in the circulating blood (up to 25× more). In regard to capillaries in the lungs, brain, liver and kidneys had relatively high numbers of free parasites. This suggests that these parasites had locally escaped from the infected red blood cells, where they could potentially invade new red blood cells. As *B. canis* divides by binary fashion (Figure 6), it is expected that the number of parasites in an erythrocyte to be 2^n^ in which n is an integer number (2^0^ = 1, 2^1^ = 2, 2^2^ = 4, etc.). However, because infected erythrocytes are trapped in fibrin deposits, there is a reasonable chance that such free parasites could infect an erythrocyte that is already infected. This could explain why erythrocytes that contain odd numbers of parasites (e.g., 3, 5, 7) are found. Moreover, the fact that the percentage of infected erythrocytes that contain odd numbers of parasites is higher in the capillaries than in the circulating blood (Table 1), further corroborates the idea that local proliferation occurs in infected capillaries. 

Additional evidence of local proliferation comes from malaria research. A detailed study on sequestration in *P. falciparum* malaria showed that within the same brain, some (parts of) capillaries appeared to be blocked by schizonts, whereas other capillaries appeared blocked with erythrocytes that contained relatively young ring forms of *P. falciparum* [32]. Although it could be hypothesized that the ring-infected erythrocytes display receptors that bind to yet other ligands on endothelial cells than schizont-infected erythrocytes, the simpler explanation is that at least a number of schizont-infected erythrocytes that sequester become localized and continue to proliferate. A single *P. falciparum* schizont results in 12–24 merozoites, which in the presence of sufficient uninfected red blood cells, can lead to a relatively synchronous local infections as seen in the studies presented by Silamut et al. [32].

## 10. Role of Soluble Proteins from *B. canis*-Infected Erythrocytes

It is still not entirely clear what triggers the inflammatory response when dogs are infected with *B. canis*. As dogs can be successfully vaccinated against the clinical signs of *B. canis* infection using soluble parasite antigens from in vitro cultures (SPA; [15]), and because the effect of vaccination was not always related to an effect on parasitaemia but on the amount of SPA in plasma after challenge infection [33], it was hypothesized that these soluble antigens triggered inflammatory responses (Figure 14). Vaccinated dogs would then produce neutralizing antibodies against SPA and as such, limit inflammation and resulting pathology. In order to determine whether SPA induced the acute phase response, a lysate prepared from *B. canis*-infected erythrocytes was infused into dogs. The amount of SPA that was infused was such that similar concentrations in plasma were attained as that found in experimentally infected dogs. The results showed that the acute phase response (plasma CRP levels) was triggered within hours after infusion [34]. This suggests that during parasite proliferation when *Babesia* merozoites escape from the red blood cell, proinflammatory responses are triggered. Importantly, increased levels of CRP in plasma (although lower) were also found upon infusion of lysates from normal erythrocytes, hence at least part of this trigger comes from self-antigens from the red blood cell stroma. The importance of this finding is that once infected erythrocytes are localized at a spot, proliferation releases normal red blood cell components that perpetuate the acute phase response and stimulate coagulation. These reactions consolidate parasite localization. Given the fact that *B. canis*-infected erythrocytes localize in the microvasculature, the number of infected erythrocytes in the circulating blood might not be an accurate measure of the parasite load. The possibility remains that vaccination against SPA limits parasite proliferation which would result in less SPA being produced, in which case immunity would be anti-parasite, and not anti-disease, as hypothesized before. 

## Figures and Tables

**Figure 1 pathogens-08-00094-f001:**
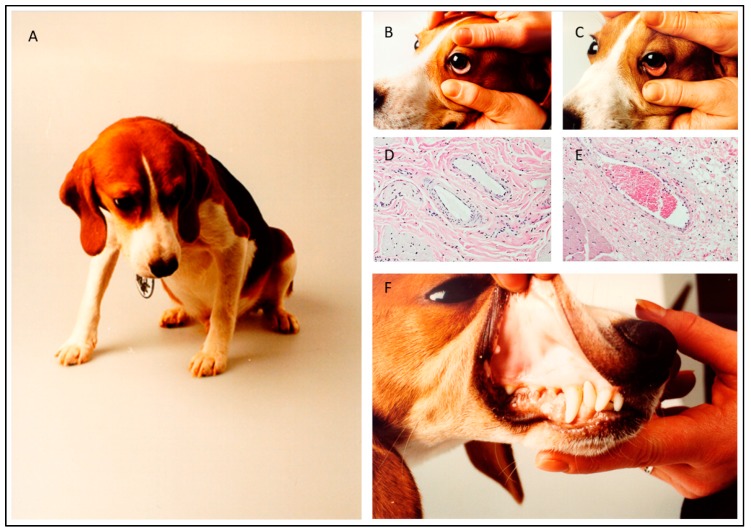
Clinical appearance of a dog that was infected with 1-2 × 10^6^
*B. canis*-infected erythrocytes from an infected donor dog. Habitus (**A**), pale-white mucosae (**B** and **F**). Normal appearance of mucosae of the eyelid (**C**). Histology shows the mucosae of the lip of an infected (**D**) and a normal dog (**E**). Note empty capillaries in mucosae of the infected dog (D).

**Figure 2 pathogens-08-00094-f002:**
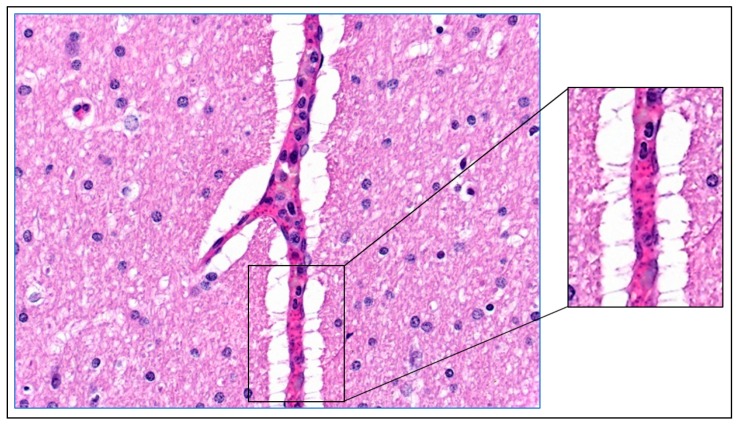
Histological section of brain tissue of a dog that developed severe clinical signs upon infection with *B. canis-*infected erythrocytes. The capillary appears entirely blocked and surrounded with perivascular oedema. The insert shows a part of the capillary with high numbers of parasites (small dots). HE-stain.

**Figure 3 pathogens-08-00094-f003:**
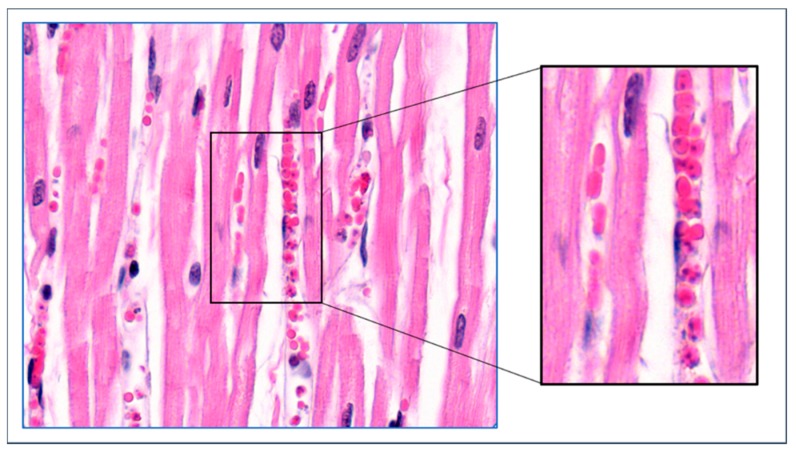
Obstructed capillary filled with *B. canis*-infected erythrocytes. The capillary in the right part of this section of heart tissue of a *B. canis*-infected dog is completely filled with infected erythrocytes. The adjacent capillary at the left appears not affected. HE stain.

**Figure 4 pathogens-08-00094-f004:**
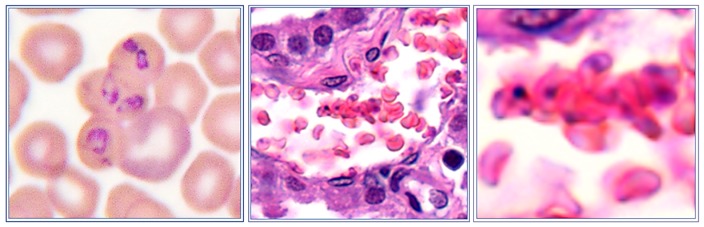
Agglutinates infected erythrocytes from a dog infected with *B. canis*. The small agglutinates are found on smears from blood taken from the jugular vein (left panel) and in capillaries of internal organs (choroid plexus). Almost all erythrocytes of these agglutinates are infected (right panel). HE stain.

**Figure 5 pathogens-08-00094-f005:**
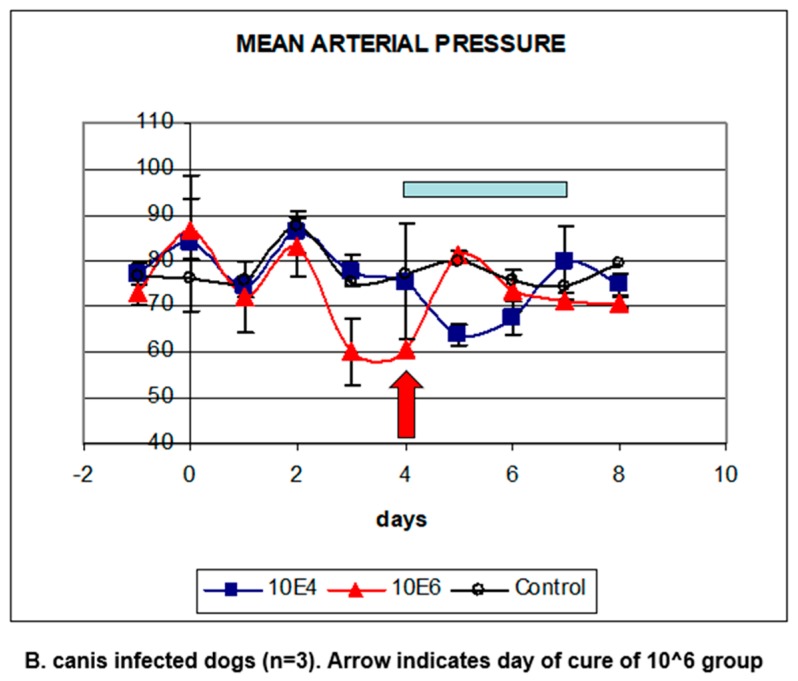
Blood pressure in *B. canis*-infected dogs. The groups of three dogs were infected with 1 × 10^6^ (red triangles) or 1 × 10^4^
*B. canis*-infected erythrocytes. Both groups of dogs developed hypotension (expressed as mean arterial pressure, MAP). Dogs infected with the highest dose were cured chemotherapeutically on day 4 after infection (arrow), the group of dogs infected with the lowest dose was cured at the end of the observation period after the phase of hypotension (blue bar).

**Figure 6 pathogens-08-00094-f006:**
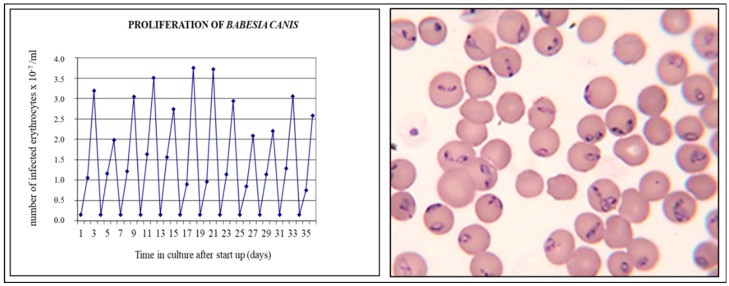
Proliferation of *B. canis* in continuous *in vitro* culture [4]. The left panel shows the development of parasitaemia in continuous *in vitro* cultures that were diluted every 48 h with normal erythrocytes after which cultivation was resumed. The parasites divide by binary fission (**right panel**), and the multiplication factor in the first 24hrs is approximately 10 (**left panel**).

**Figure 7 pathogens-08-00094-f007:**
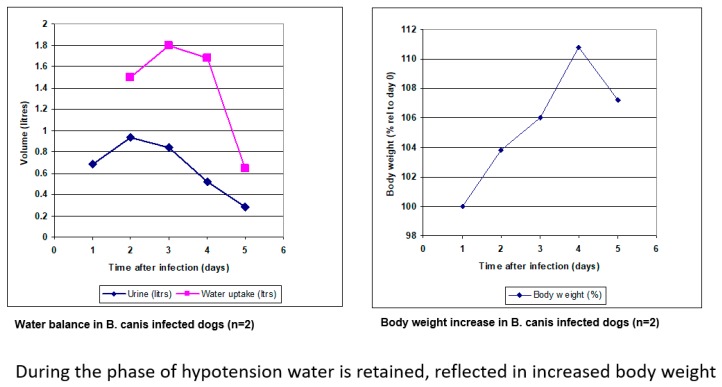
Water balance in *B. canis*-infected dogs. Dogs retain water through increased water intake (drinking) and reduced urination (**left panel**), which is reflected in an increase of body weight (**right panel**).

**Figure 8 pathogens-08-00094-f008:**
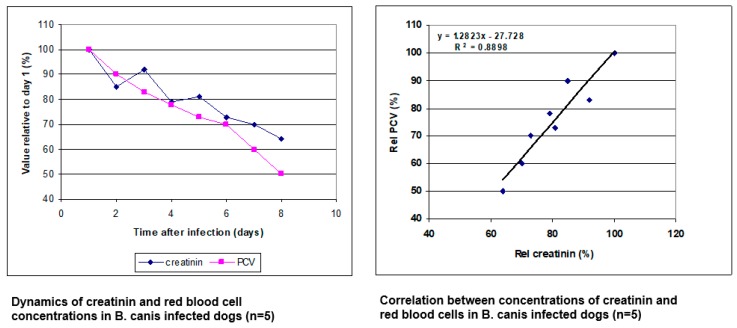
The dynamics of erythrocyte concentration and creatinin concentration in dogs that had been infected with 2x10^6^
*B. canis*-infected erythrocytes. The data represent the concentration relative to the value at the day of infection (**left panel**). The fact that both concentrations show a positive linear relationship indicates that the blood was diluted (**right panel**).

**Figure 9 pathogens-08-00094-f009:**
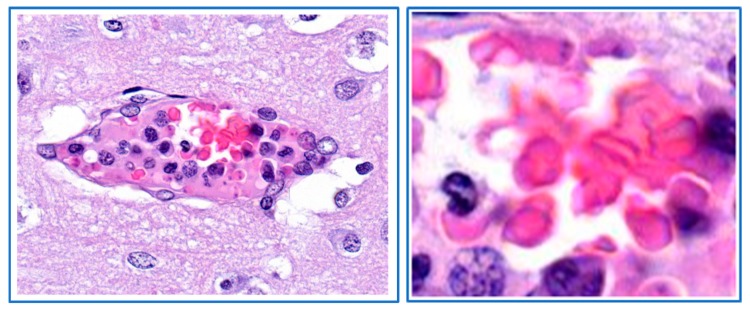
Histological section of brain tissue of a dog that developed severe clinical signs upon infection with *B. canis-*infected erythrocytes. The deposits of fibrin (amorphous pink staining material) with leukocytes are found at the margin of the capillary (**left panel**). In the central part of the capillary, normal red blood cells are found (**right panel**). HE stain.

**Figure 10 pathogens-08-00094-f010:**
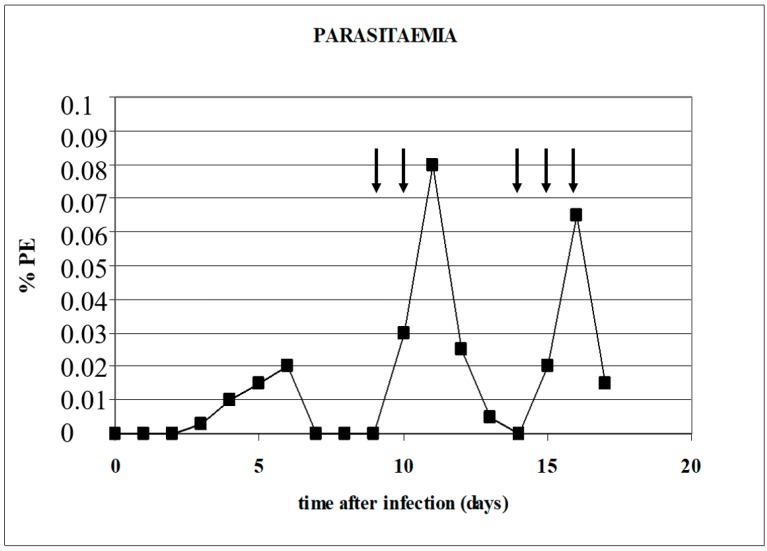
The effect of a heparin injection on the parasitaemia of a *B. canis*-infected dog. The time points of heparin administration are indicated with arrows [23].

**Figure 11 pathogens-08-00094-f011:**
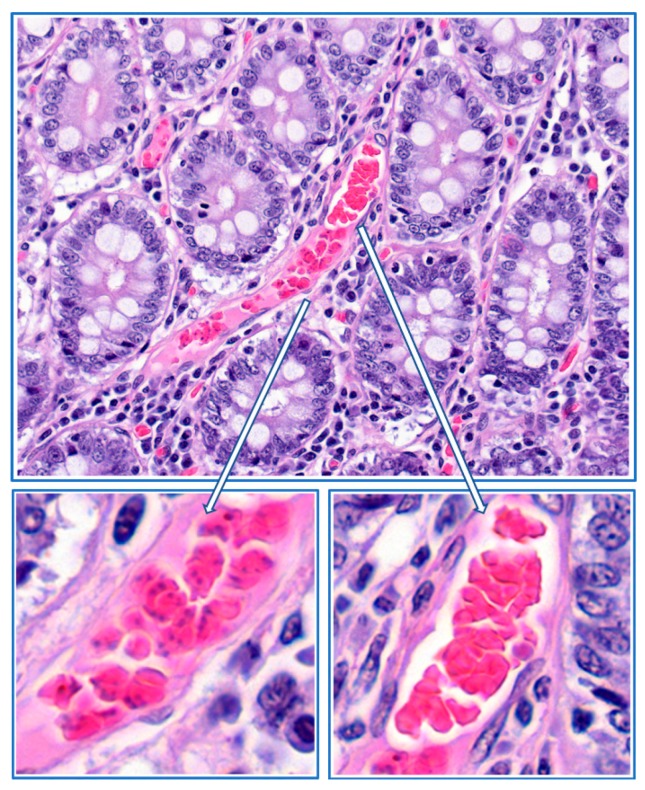
Histological section of the gut tissue of a dog that developed severe clinical signs upon infection with *B. canis-*infected erythrocytes. The deposits of fibrin (amorphous pink staining material) with infected erythrocytes are found at the margin of the capillary (**lower left panel**). In the central part of the capillary, normal red blood cells are found (**lower right panel**). HE stain.

**Figure 12 pathogens-08-00094-f012:**
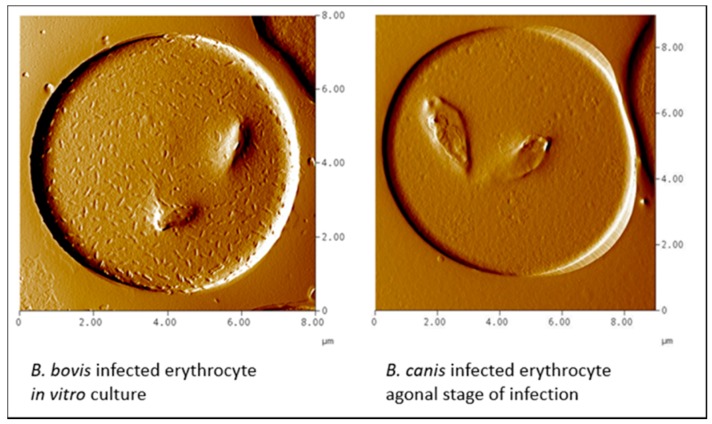
Atomic force microscopy of *B. bovis*-infected erythrocyte (**left**) and a *B. canis*-infected erythrocyte (**right**). The latter was isolated from blood of a dog in the agonal phase of infection that was cultured *in vitro* for 24 h to increase parasitaemia. Note the presence of knob-like structures on the *B. bovis*-infected erythrocyte (Courtesy of Prof. Brian Cooke).

**Figure 13 pathogens-08-00094-f013:**
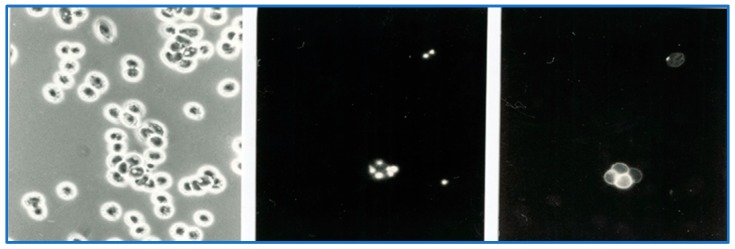
Parasite antigens at the surface of *Babesia canis*-infected erythrocytes. Blood smear of *B. canis*-infected erythrocytes observed with bright-field microscopy (**left panel**), fluorescence microscopy that shows parasite nuclei when stained with Hoechst stain (centre panel), and live immunofluorescence of *B. canis*-infected erythrocytes stained with FITC-labelled antibodies against dog immunoglobulins after incubation with serum from dogs that were immune to *B. canis* (**right panel**).

**Figure 14 pathogens-08-00094-f014:**
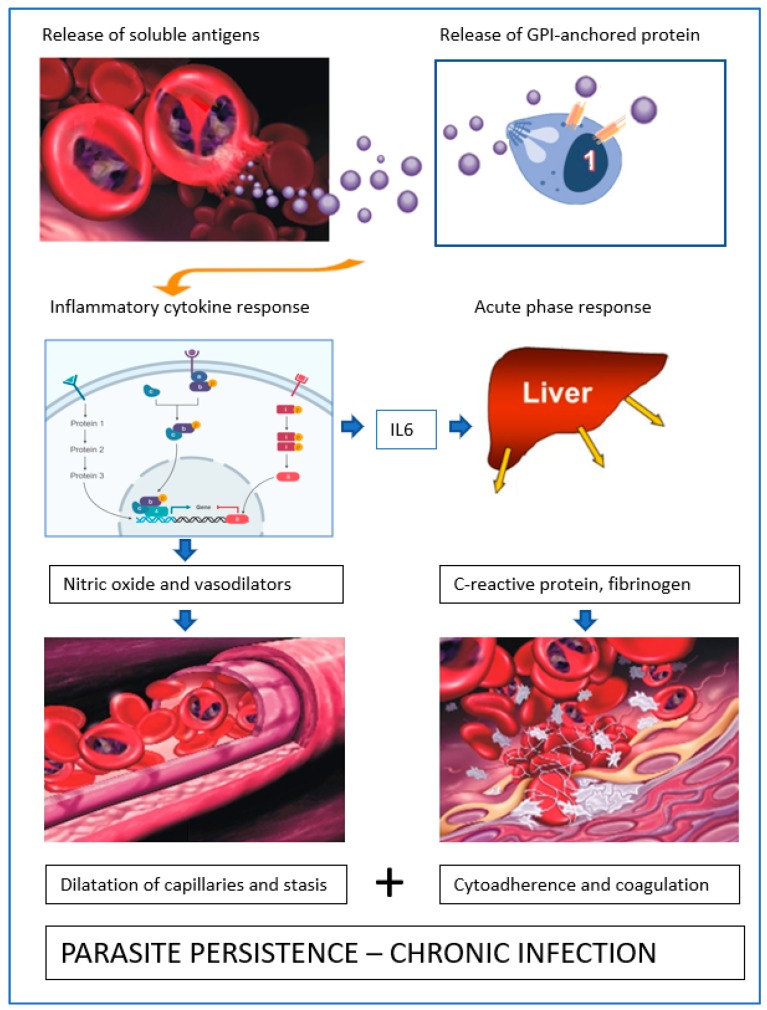
Schematic representation of key events leading to parasite persistence. The soluble antigens excreted by *Babesia canis* in erythrocytes and/or released from the parasite surface, provoke an inflammatory response that leads to the generation of vasodilatory compounds. As a result, the capillaries are dilated and the blood flow is decreased. In addition, the pro-inflammatory cytokine IL-6 is produced that stimulates liver cells to produce acute phase proteins, such as fibrinogen. Due to reduced blood flow, the propensity of infected erythrocytes to adhere to the capillary wall is increased. The activation of the coagulation system leads to conversion of fibrinogen to fibrin and the formation of blood clots with entrapped infected erythrocytes. Parasites proliferate in these clots, thus perpetuating the infection.

**Table 1 pathogens-08-00094-t001:** The evaluation of the percentage of infected erythrocytes in the blood and in internal organs of dogs that succumbed to babesiosis. In addition, the relative distribution of erythrocytes with a specified number of parasites (range 1–16 *Babesia* parasites per erythrocyte) is presented as a percentage. The data are compiled from Graham-Smith 1905. PE-high is the highest parasitaemia that was found; PE-low is the lowest parasitaemia that was found; #Free Par to PE is the number of free *Babesia* parasites per infected erythrocyte (PE); #Par/RBC is the number of *Babesia* parasites that was detected in an infected red blood cell (RBC)**.**

Parasite Distribution in Dogs That Succumbed To Babesiosis (Adapted From Graham-Smith 1905)													
	Peripheral Blood	Brain Capillaries	Spinal cord Vessels	Lungs Capillaries	Heart Capillaries	Liver Capillaries	Spleen Pulp	Capillaries	Kidney Capillaries	Adrenals Capillaries	Pancreas Capillaries	Lymph nodes Capillaries	Marrow femur
Dilatation		+	+	++	++	+++		yes	++	+	+++	+	na
PE-high	6%	88%	neg	52%	72%	53%	12%	48%	95%	70%	Numerous		53.5%
PE-low	0.10%	nd	neg	4	nd	23%	3.7%	nd	nd	nd	nd		2.0%
# Free Par to PE	0.06	0.50	na	1.50	nd	0.40		0.11	0.45	0.21	0.11		0.13
#Par/RBC	%	%		%		%		%	%	%	%	%	%
1	67.76	38.33		38.97		40.84		46.27	48.37	34.33	50.93	40.03	29.59
2	28.66	39.97		52.37		50.08		42.06	43.52	52.00	44.14	46.04	52.06
3	0.71	2.02		1.06		1.25		1.14	1.89	1.71	0.53	0.32	1.94
4	2.54	14.75		6.70		6.81		8.79	5.23	10.01	4.12	8.86	12.39
5	0.04	0.25		0.07		0.19		0.26	0.15	0.21		0.16	0.40
6	0.12	1.38		0.31		0.22		0.48	0.39	0.78	0.16	0.94	1.19
7				0.03		0.05		0.02	0.03			0.16	0.18
8	0.12	2.52		0.31		0.47		0.84	0.36	0.85	0.10	2.69	1.80
9	0.01												0.03
10	0.01			0.10		0.04		0.09		0.07		0.31	0.11
11													
12	0.01	0.25				0.02						0.16	0.02
13													0.01
14		0.13		0.07				0.02					0.01
15		0.13											
16	0.01	0.25				0.02		0.02				0.32	0.15
Total COUNTED	22,589	793		2910		5345		5347	5792	1398	1869	632	8660

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
