# Peer review of "Mechanisms Involved in the Persistence of *Babesia canis* Infection in Dogs"

_pathogens, 2019, doi:10.3390/pathogens8030094_

Round 1
Reviewer 1 Report
The observation of differential accumulation of Babesia canis-infected erythrocytes in tissues compared to the peripheral circulation provides a clear opportunity to draw parallels with the seminal findings of Marchiafava and Bignami in malaria and their hypothesis of sequestration of infected forms. This paper brings together a range of observations to build a case for the capture of infected erythrocytes in the microvasculature and anchoring in fibrin networks. This hypothesis is an attractive one and is largely consistent with the observations provided, but in places there is a lack of definitive evidence.
Comments
1. In the abstract the author talks about the destruction of the infected erythrocyte (at schizogony) making it difficult to explain the “further accumulation of infected erythrocytes”. Why? The surface of the microvasculature is large so can probably accommodate rounds of parasite accumulation, and debris is continuously removed by monocyte surveillance. In structural terms for the paper, this issue is not covered in the main text.
2. Although the references to the various panels in Figure 1 are reasonably clear, it might help to add labels to each panel so they can be referenced more simply.
3. The language used to describe the dilatation in Table 1 is a bit confusing. Could it be replaced with a semi-quantitative scoring system (e.g. ‘++’ and ‘++++’).
4. The issue of agglutinate formation with parasitaemias of <1% (lines 108-113) is not as unlikely as has been suggested and so the premise that they need to be formed “locally before entering the blood circulation” is possible but not necessary. No data are presented to support a mechanism for agglutination.
5. It was not clear to me how the left panel in Figure 6 shows a proliferation factor of 11. Why does the number of infected red blood cells crash every 3-4 days? Some more explanation of this figure would be helpful.
6. The comparison with hypotension in malaria is not particularly helpful to this review and is not a major issue in malaria infection. I would suggest starting section 6 at line 157 to talk directly about babesiosis. Similarly at lines 197-200, the issues in malaria infection are multi-factorial and so to focus on venous pooling is not helpful.
7. As with all correlations, the observation that as parasitaemia increases, plasma fibrinogen levels decrease, is an association but does not show causality (lines 210-211). This is an interesting hypothesis and certainly worth investigation, but the evidence in this review is quite limited:
a. It is difficult to see the fibrin deposits in Figure 9 (and Figure 11) and no formal quantitation is provided.
b. There could be a number of reasons for the release of infected erythrocytes on heparin treatment (note the reference to this work on line 221 is incorrectly labelled as “figure 9”). A non-anti-coagulant GAG would be a useful control for this experiment. Were the fibrin nets disrupted after treatment?
c. Why would infected erythrocytes be preferentially released (line 224). I think I understand what is being said, namely that if the sequestered masses are broken down by heparin, the release of the cells held there will have predominantly infected erythrocytes, hence the rise in peripheral parasitaemia. This could be said more clearly.
d. How can you discriminate between fibrin deposits trapping (line 243) parasites or the parasites being co-located with fibrin deposition, possibly caused by the parasites being attached to the endothelium?
8. There is a minor typo on line 233 (associate to associated).
9. The concept of ‘local proliferation’ is a difficult one:
a. The observation that most of the cells in the fibrin mass are infected could be due to the perturbation of the capillary flow by the accumulation of infected forms, so that once the flow has been disrupted, the chance that other infected erythrocytes will be recruited will be higher. This has been formally demonstrated for leukocyte recruitment and is presumed to act for sequestration foci in Plasmodium falciparum malaria.
b. Can erythrocytes that have already been infected by Babesia support subsequent invasion? This would seem to be a poor strategy by the parasite, as the less mature forms would be destroyed on schizogony of the first wave of invaders. Is it more likely that the distribution of multiply infected erythrocytes is due to preferential invasion – possibly due to different amounts of host receptors being available on the red cell surfaces.
c. Local proliferation is not thought to take place in P. falciparum malaria (lines 321-327). Re-invasion takes place in the sequestration foci but the newly infected erythrocytes are released into the peripheral circulation and are found there up to 16 hours post-invasion, at which point the cytoadherence receptors are available on the infected red blood cell membrane. This is clearly seen in the kinetics of peripheral parasitaemia.
10. In the legend to Figure 14, when capillaries are dilated, blood flow is increased, not decreased (line 362). Also, how is the coagulation system activated (line 365)?
Several parallels are made in this review to malaria, where activation of the coagulation system through sequestration has become big news. The mechanisms for this are becoming increasingly understood from careful clinical and molecular studies. The possibility that Babesia performs a similar role at the endothelial surface is an attractive one, but the evidence supporting this is very limited. Thus, this review presents some intriguing suggestions but little definitive evidence and does not really support any mechanisms. As a call for more work to be done, it is definitely supportable.
Reviewer 2 Report
This is a well-written review paper about the mechanisms of persistence of Babesia canis in affected dogs.
The only remarks are referred to proper name Babesia (line 26 and throughout all the text). Please write it in italics, as well as the term "in vitro"
